Resource

# Identification of biomarkers, pathways, and therapeutic targets for EGFR–TKI resistance in NSCLC

Leilei Zhu[1] , Shanshan Gao[1], Xianya Zhao[1], Ying Wang[2]

**This study aimed to map the hub genes and potential pathways that might be involved in the molecular pathogenesis of EGFR–TKI resistance in NSCLC. We performed bioinformatics analysis to identify differentially expressed genes, their function, gene interactions, and pathway analysis between EGFR–TKI-sensitive and EGFR–TKI-resistant patient-derived xenotransplantation samples based on Gene Expression Omnibus database. Survival analysis was performed via the GEPIA database (GEO). The relationship between the key gene ITGAM and the therapeutic candidates was retrieved from DGIdb. A total of 1,302 differentially expressed genes were identified based on GEO. The PPI network highlighted 10 potential hub genes. Only ITGAM was linked to poor DSF in NSCLC patients. A total of 10 drugs were predicted to be potential therapeutics for NSCLC with EGFR–TKI resistance. This study indicates the hub genes related to EGFR–TKI resistance in NSCLC through bioinformatics technologies which can improve the understanding of the mechanisms of EGFR–TKI resistance and provide novel insights into therapeutics.**

## Introduction

Lung cancer is the second most commonly diagnosed cancer worldwide and the leading cause of cancer death. Non-small-cell lung cancer (NSCLC) accounts for 85% of all lung cancers (1, 2). Targeted drugs represented by epidermal growth factor receptor (EGFR)–tyrosine kinase inhibitors (TKIs) have brought revolutionary progress in the treatment of advanced NSCLC (3, 4). With the extensive clinical application of EGFR–TKIs, acquired resistance has become a challenge faced by clinicians (5, 6). Despite extensive endeavors to fathom the molecular underpinnings of EGFR–TKI-acquired resistance, a comprehensive understanding of the underlying molecular mechanisms and pivotal genes remains elusive (7, 8).

In tandem with the rapid evolution of gene sequencing and bioinformatics analysis technologies, researchers now wield the capacity to access high-throughput microarray data and next-generation sequencing functional genomics data through international repositories such as the Gene Expression Omnibus (GEO) and The Cancer Genome Atlas (TCGA) (9, 10, 11). These online repositories afford simultaneous access to expression data for a multitude of genes, which can be meticulously analyzed to unearth prospective biomarkers and therapeutic targets implicated in EGFR–TKI drug resistance in NSCLC. Nevertheless, the identification of these markers predominantly hinges on the comparison of normal and cancerous tissue samples, with an added emphasis on obtaining drug-resistant samples—a formidable challenge in itself. Notably, most of the sequencing studies have thus far concentrated on artificially induced drug-resistant cell lines, which have inherent limitations in dissecting the key genes underpinning drug resistance in tumors. Patient-derived xenotransplantation (PDX) model—a potent tool in the realm of cancer biology research, distinguished by its aptitude for preserving the salient attributes of patient tumors. Consequently, it offers superior suitability for experimental inquiries into the molecular mechanics of tumor progression and drug resistance (12, 13).

In this study, we embarked on the pursuit of identifying differentially expressed genes (DEGs) between EGFR–TKI-sensitive and acquired drug-resistant NSCLC xenograft tumor samples. This endeavor was realized through a meticulous mining of the gene expression microarray datasets GSE64472 and GSE130160, subsequently subjecting the DEGs to Gene Ontology (GO) annotation and Kyoto Encyclopedia of Genes and Genomes (KEGG) pathway analysis. This analytical journey was facilitated by the Database for Annotation, Visualization, and Integrated Discovery (DAVID) online tool. Furthermore, we crafted a protein–protein interaction (PPI) network using the Search Tool for the Retrieval of Interacting Genes (STRING) database, followed by in-depth analysis using Cytoscape software, culminating in the identification of hub genes. Complementing these efforts, we undertook a survival analysis of patients displaying aberrant hub gene expression, drawing upon data gleaned from TCGA database. The insights gleaned from this comprehensive endeavor are poised to illuminate the roles played by these genes in the genesis of EGFR–TKI resistance. This study thus represents a pivotal contribution to the understanding of the molecular mechanisms underpinning EGFR–TKI resistance,

[1]Department of Anesthesiology, The First Affiliated Hospital of Anhui Medical University; Anhui Public Health Clinical Center, Hefei, China  [2]Department of Respiratory Medicine, Anhui Provincial Children's Hospital (Children's Hospital of Fudan University Anhui Hospital), Hefei, China

Correspondence: etyywangying@163.com

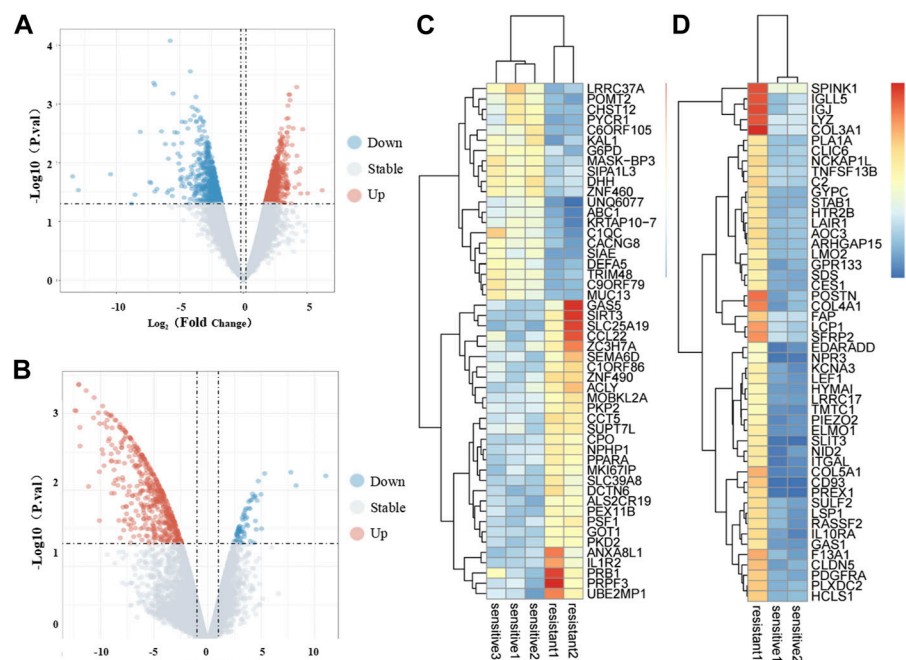

**Figure 1. Identification of differentially expressed genes.**
**(A, B)** Volcano plot of DEGs in the GSE64472 and GSE130160 datasets. The red dots represent up-regulated genes, the green dots represent down-regulated genes, and the black dots represent genes with no significant difference in expression. **(C, D)** Heatmap of the top 50 DEGs in the GSE64472 and GSE130160 datasets. Red represents up-regulated genes, and blue represents down-regulated genes.

whereas also unveiling novel gene targets that hold immense promise for future investigations.

# Results

## Identification of DEGs

To identify meaningful biomarkers distinguishing the EGFR–TKI-sensitive and -resistant groups, we used the R limma package and applied the criteria of $P < 0.05$ and $|logFC| > 2$. This analysis revealed a total of 1,302 DEGs when comparing the EGFR–TKI-resistant and -sensitive groups using the GSE64472 and GSE130160 datasets. Specifically, GSE64472 yielded 775 DEGs in the resistant group, comprising 339 up-regulated and 436 down-regulated DEGs. In contrast, GSE130160 produced 529 DEGs, with 479 up-regulated and 52 down-regulated DEGs in the resistant group. The volcano plot depicting DEGs is presented in Fig 1A and B, whereas Fig 1C and D illustrate the expression heatmap of the top 50 DEGs, sorted by Padj value.

## GO and KEGG enrichment analyses of DEGs

To elucidate the functional roles of these DEGs in EGFR–TKI resistance progression in NSCLC, we conducted functional predictions. This entailed GO analysis encompassing cellular components (CC), molecular function (MF), and biological processes (BP), and KEGG analysis, both using the DAVID database. A false discovery rate–corrected $P$-value < 0.05 and an enrichment score > 1.5 were adopted as the significance thresholds for GO functional enrichment analysis, resulting in the mapping of 1,302 DEGs into 48 significantly enriched functional clusters. Among these, 11 GO terms were significantly enriched in cellular components, including "plasma membrane,"

"extracellular region," "extracellular space," "extracellular exosome," and "proteinaceous extracellular matrix" (Fig 2A). Molecular function analysis (Fig 2B) revealed enrichment in 8 GO terms, such as "integrin binding," "cytokine activity," "growth factor activity," "protein homo-dimerization activity," and others. Moreover, biological processes (Fig 2C) yielded a total of 29 enriched terms, predominantly encompassing "immune response," "cell adhesion," "extracellular matrix organization," and more. KEGG analysis integrated 1,303 DEGs into 16 enriched functional clusters, including pathways such as "cytokine–cytokine receptor interaction," "melanogenesis," "circadian entrainment," "basal cell carcinoma," and "dopaminergic synapse" (Fig 2D).

## Integration of the PPI network and module analysis

We constructed and visualized the PPI network of the 1,302 DEGs using the STRING database. Subsequently, we pruned isolated nodes and loosely connected gene nodes, resulting in a complex multicenter interaction network with 1,402 nodes and 4,761 edges (Fig 3A). The average node degree was 7.92, and the average local clustering coefficient was 0.286. Among the 1,402 nodes, the top 20 DEGs with the highest node degrees were screened (Fig 3B and C). Expression patterns of the top 20 genes across GSE64472 and GSE130160 samples are illustrated in Fig 3D and E. The top 10 DEGs identified were IL6, IL10, CXCL9, ITGAM, CCL5, CD4, IDO1, HAVCR2, TLR9, and CCR7. Detailed information regarding these hub genes, including their full names and functions, can be found in Table 1.

## Disease-free survival (DFS) analyses of hub genes in NSCLC

In an effort to identify genes among the hub genes that could potentially contribute to EGFR–TKI resistance and serve as predictors of cancer progression, we conducted DFS analysis for NSCLC

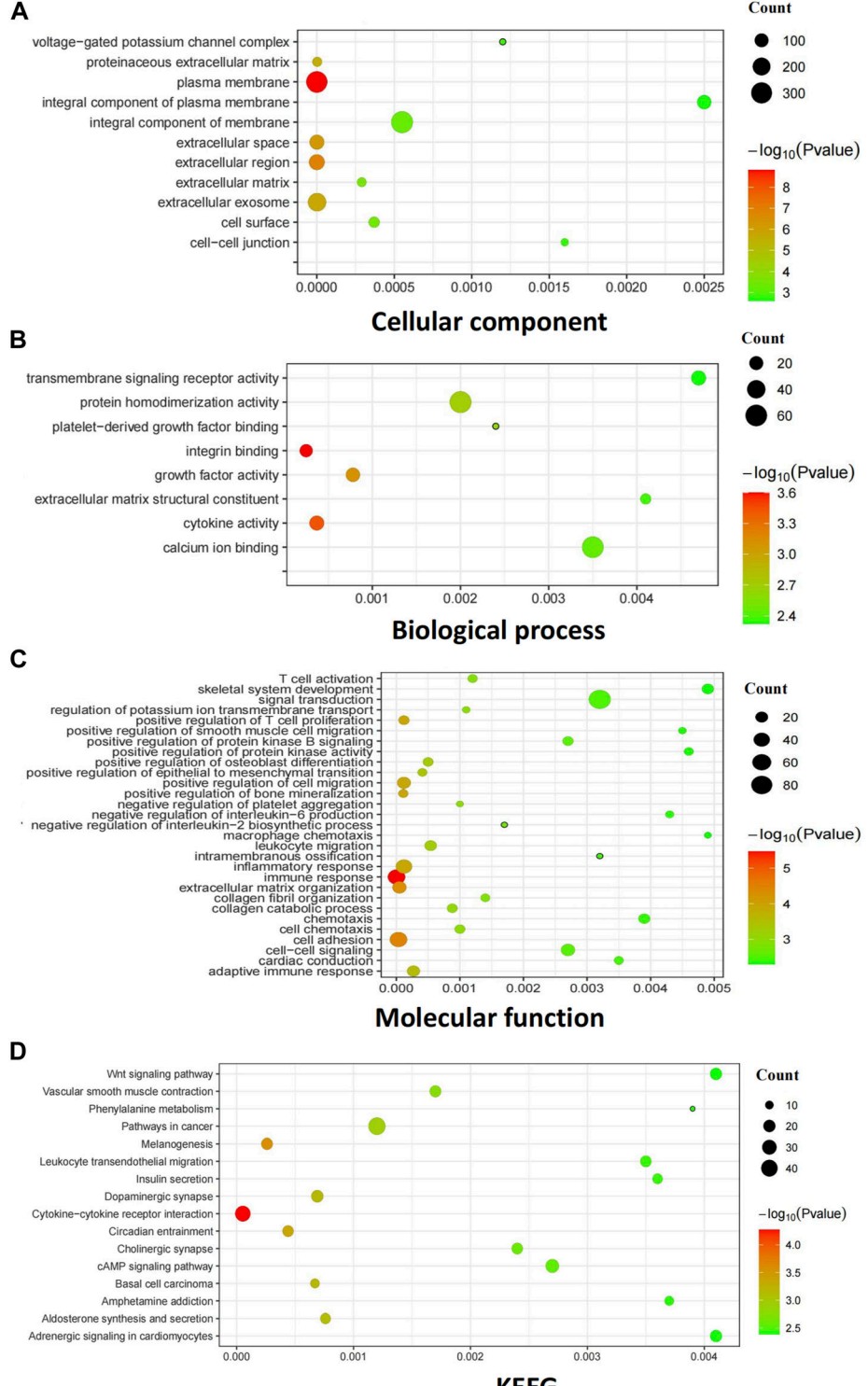

**Figure 2. Gene Ontology and KEGG pathway analysis of DEGs in NSCLC.**
**(A)** GO covering the domains of molecular functions (MF). **(B)** GO covering the domains of biological processes (BP). **(C)** GO covering the domains of cellular components (CC). **(D)** KEGG pathways that were the most significantly up-regulated pathways during SCLC. The bubbles represent the enrichment pathway with *P*-values < 0.05. The bubble size represents the number of enriched target genes in the process. The bubble color represents –log₁₀ (*P*-value); from green to red, the *P*-value decreases. The Y-axis represents the enrichment target of GO or pathway. The X-axis is the RichFactor: its counts divided by the third column.

patients using the Kaplan–Meier plotter database. Among these genes, elevated ITGAM expression was found to be associated with improved NSCLC patient DFS (HR = 0.73, 95% CI: 1.26–1.81, *P* = 0.045) (Fig 4). These results underscore the central role of ITGAM in the context of EGFR–TKI resistance.

## Drug interaction prediction for EGFR–TKI resistance

The relationship between the EGFR–TKI resistance–specific gene ITGAM and the corresponding potential therapeutic candidates was retrieved from DGIdb. A total of 207 drugs were predicted to

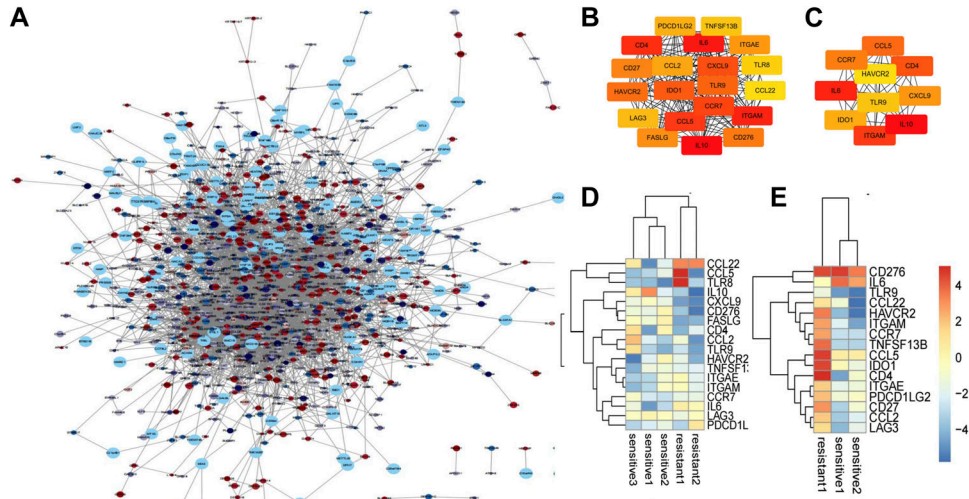

**Figure 3. PPI analysis of DEGs based on Cytoscape.**
**(A)** Visualized PPI analysis of DEGs. **(B)** Top 20 genes with the highest MCC scores in DEGs. **(C)** Top 10 genes with the highest MCC scores in DEGs; a darker color represents higher MCC scores. **(D)** Heatmap of the top 20 DEGs in GSE64472. **(E)** Heatmap of the top 20 DEGs in GSE130160. Red represents up-regulated genes, and blue represents down-regulated genes.

interact with ITGAM, with some of the highest frequency drugs including liarozole, rovelizumab, dimethyl sulfoxide, clarithromycin, fentanyl, phenylephrine, theophylline, morphine, hydrocortisone, and atorvastatin (Table 2).

## Discussion

The mechanisms underlying EGFR–TKI resistance in NSCLC can be broadly categorized into acquired resistance following EGFR–TKI treatment and primary resistance characterized by cancer cells relying on alternative oncogenes like KRAS. The most prevalent mechanism of acquired resistance involves the emergence of an EGFR T790M gatekeeper mutation, occurring in a substantial percentage of cases (4~50%) (14, 15, 16, 17). Other reported mechanisms encompass MET amplification (18), hepatocyte growth factor expression (19), and epithelial–mesenchymal transition (20). Thus, it is imperative to decipher the molecular mechanisms driving EGFR–TKI resistance in NSCLC to identify novel therapeutic targets for future interventions.

Cancer cell lines have long been indispensable for drug screening; however, they have been cultured for numerous generations and deviate significantly from primary tumor tissues in terms of genetic makeup and behavior (21). Subcutaneous or orthotopic cell-derived tumor xenograft models (CDX models) inadequately replicate the genetic diversity observed in human tumors, which has resulted in low clinical trial response rates despite effective outcomes in traditional animal models (CDX models) (22). Patient-derived xenograft models (PDX models) have emerged as a promising alternative in preclinical cancer research over recent years. These models have demonstrated the preservation of genetic characteristics compared with primary human tumor tissue (12, 13, 23, 24). For instance, PDX models have successfully correlated gemcitabine responses in pancreatic ductal adenocarcinoma with clinical patient outcomes (25), and the efficacy of sorafenib in hepatocellular carcinoma PDX models closely mirrors patient responses (26, 27). Thus, PDX models offer a valuable platform for identifying molecular biomarkers associated with drug sensitivity or resistance and assessing the efficacy of novel drugs.

Microarray technology is a cornerstone in exploring gene expression patterns in complex disorders (11). However, prior bioinformatics studies often focused on results derived from cell line–based or CDX models. In our study, we overcame this limitation by identifying two microarray datasets associated with acquired resistance in vivo. These datasets (GSE64472 and GSE130160) contained EGFR–TKI-sensitive PDXs initially responsive to VEGFR inhibition but subsequently developing resistance following prolonged vandetanib and osimertinib treatment. Using these datasets, we conducted comprehensive bioinformatic analyses comparing gene expression in EGFR–TKI-sensitive and -resistant PDX samples. Our objective was to identify and functionally characterize hub genes involved in EGFR–TKI resistance, providing insights into the molecular mechanisms driving drug resistance and offering novel gene targets for future studies.

In this study, we performed an intersection analysis of the two datasets to enhance the reliability of our identified DEGs. Ultimately, 1,203 DEGs were uncovered. GO functional analysis revealed their enrichment in critical categories, including "plasma membrane," "extracellular region," "integrin binding," "cytokine activity," "growth factor activity," "protein homodimerization activity," "platelet-derived growth factor binding," "immune response," "cell adhesion," and "extracellular matrix organization." Notably, previous studies have highlighted the up-regulation of integrin $\beta3$ post-EGFR–TKI treatment, underscoring its role in EGFR–TKI resistance (28, 29, 30, 31). Furthermore, KEGG pathway analysis demonstrated the involvement of these DEGs in pivotal pathways like "cytokine–cytokine receptor interaction," "melanogenesis," and "basal cell carcinoma." Intriguingly, cytokine–cytokine receptor interactions have been identified as primary drivers of EGFR–TKI resistance in NSCLC (32). The frequent observation of the loss of microphthalmia-associated transcription factor in acquired resistance, leading to a mesenchymal-like invasive or neural crest stem cell phenotype, underscores the importance of differentiation into basal cells in drug-induced resistance (33, 34).

**Table 1. Functional roles of 10 hub genes.**

| No. | Gene symbol | Full name | Function |
| --- | --- | --- | --- |
| 1 | IL6 | Interleukin 6 | A cytokine that functions in inflammation and the maturation of B cells. |
| 2 | IL10 | Interleukin 10 | A cytokine produced primarily by monocytes and to a lesser extent by lymphocytes. This cytokine has pleiotropic effects in immunoregulation and inflammation. |
| 3 | CXCL9 | C–X–C motif chemokine ligand 9 | The protein encoded is thought to be involved in T-cell trafficking. The encoded protein binds to C–X–C motif chemokine 3 and is a chemoattractant for lymphocytes but not for neutrophils. |
| 4 | ITGAM | Integrin subunit alpha M | This gene encodes the integrin alpha M chain. This I-domain containing alpha integrin combines with the beta 2 chain (ITGB2) to form a leukocyte-specific integrin. The alpha M beta 2 integrin is important in the adherence of neutrophils and monocytes to stimulated endothelium and in the phagocytosis of complement coated particles. |
| 5 | CCL5 | C–C motif chemokine ligand 5 | This gene is one of the several chemokine genes clustered on the q-arm of chromosome 17. This chemokine, a member of the CC subfamily, functions as a chemoattractant for blood monocytes, memory T-helper cells, and eosinophils. |
| 6 | CD4 | CD4 molecule | The CD4 membrane glycoprotein acts as a coreceptor with the T-cell receptor on the T lymphocyte to recognize antigens displayed by an antigen presenting cell in the context of class II MHC molecules. |
| 7 | IDO1 | Indoleamine 2,3-dioxygenase 1 | A heme enzyme that acts on multiple tryptophan substrates. This enzyme is thought to play a role in a variety of pathophysiological processes such as antimicrobial and antitumor defense, neuropathology, immunoregulation, and antioxidant activity. |
| 8 | HAVCR2 | Hepatitis A Virus Cellular Receptor 2 | The protein belongs to the immunoglobulin superfamily, and TIM family of proteins. CD4-positive T helper lymphocytes can be divided into types 1 (Th1) and 2 (Th2) on the basis of their cytokine secretion patterns. |
| 9 | TLR9 | Toll-like receptor 9 | The protein encoded by this gene is a member of the TLR family, which plays a fundamental role in pathogen recognition and activation of innate immunity. |
| 10 | CCR7 | C–C motif chemokine receptor 7 | The protein encoded by this gene is a member of the G protein–coupled receptor family. This receptor is expressed in various lymphoid tissues and activates B and T lymphocytes. |

Through the construction of a PPI network, we identified 10 candidate hub genes (IL6, IL10, CXCL9, ITGAM, CCL5, CD4, IDO1, HAVCR2, TLR9, and CCR7) within the DEGs of our study. IL-10, an immunoregulatory component, has been implicated in promoting tumor malignancy by influencing T-cell apoptosis and tumor cell survival (35). Persistently activated IL-6/STAT3 signaling has been linked to acquired EGFR–TKI resistance in NSCLC treatment (36). CXCL9, an inflammatory chemokine, has shown inhibitory effects on NSCLC tumor growth and metastasis by reducing tumor-associated angiogenesis (37). In our study, CXCL9 exhibited higher expression in EGFR–TKI-sensitive samples, consistent with its role in inhibiting tumor-associated angiogenesis and contributing to EGFR–TKI resistance. IDO1, which is overexpressed in NSCLC, is associated with higher pathological stages and lymph node metastasis, suggesting its role in immune resistance and tumor progression (38). CCL5, known for its involvement in cancer cell migration and metastasis

(39), was also identified among the hub genes. Survival analysis revealed that only ITGAM was significantly associated with poor NSCLC patient prognosis, with elevated ITGAM expression correlating with poorer DFS. ITGAM, or CD11b, is involved in regulating macrophage polarization, proinflammatory macrophage transcription, and immune responses (40, 41, 42). Recent findings indicate that ITGAM modulates angiogenesis through cytokine expression control in murine and human cancer models (43, 44). These results suggest that ITGAM may directly or indirectly regulate EGFR–TKI resistance and could serve as a diagnostic biomarker.

In addition, we predicted drugs that could regulate the EGFR–TKI resistance-specific gene ITGAM in NSCLC patients. Among the 10 predicted drugs for ITGAM, some have demonstrated efficacy in cancer therapy or combination treatment. For instance, clarithromycin (CLM) has been shown to enhance cytotoxic effects when combined with gefitinib (GEF) in NSCLC cell lines (45). Liarozole

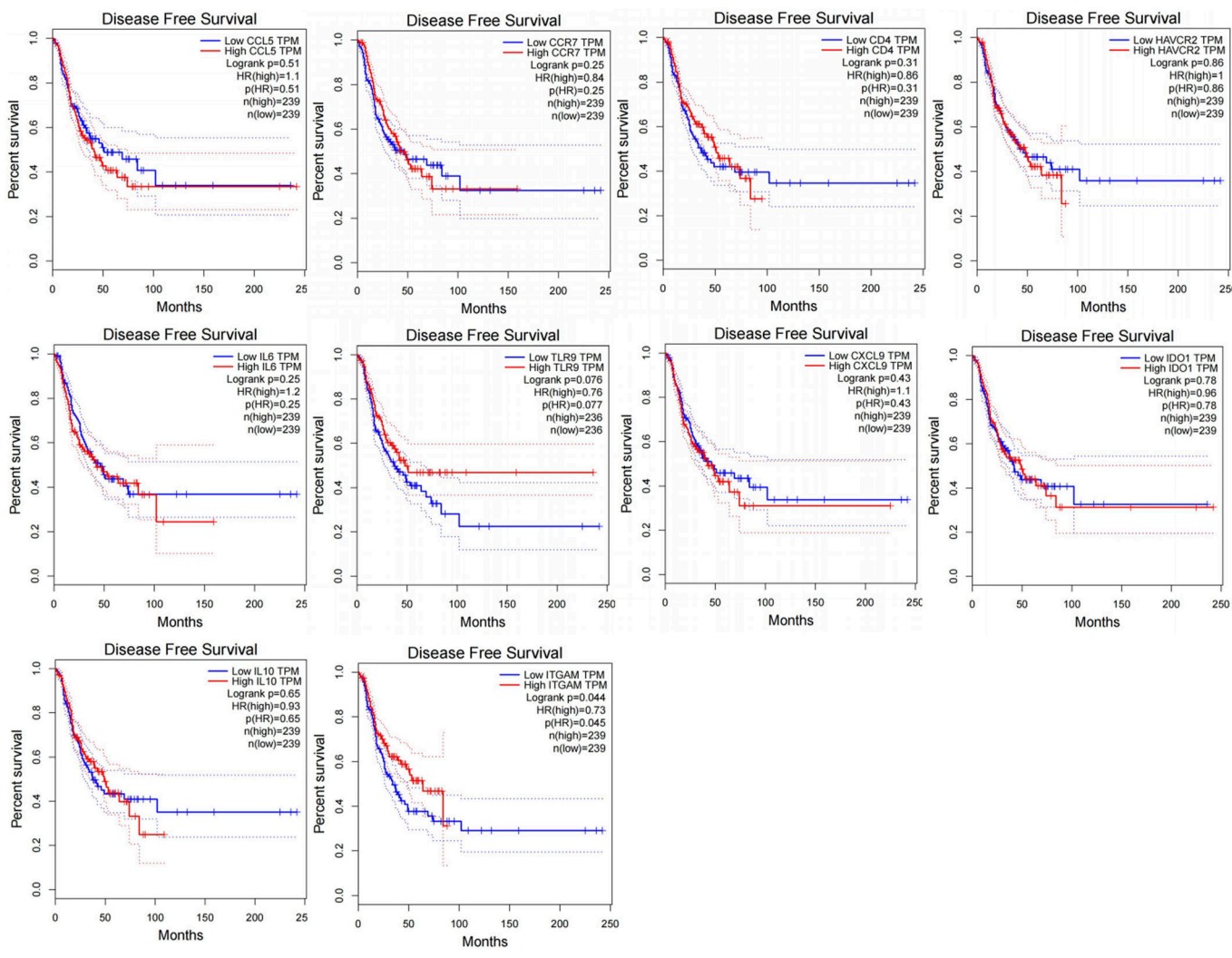

**Figure 4. Disease-free survival analyses of 10 hub genes based on The Cancer Genome Atlas.**

**Table 2. Top 10 drug predictions for the EGFR–TKI resistance–specific key gene TIMP1.**

| Drug | Interaction type and directionality | Sources | Query score | Interaction score |
|---|---|---|---|---|
| Liarozole | n/a | NCI | 2.92 | 4.25 |
| Rovelizumab | Antagonist (inhibitory) | ChemblInteractions | 1.46 | 2.13 |
| Dimethyl sulfoxide | n/a | NCI | 1.46 | 2.13 |
| Clarithromycin | n/a | NCI | 1.35 | 0.98 |
| Fentanyl | n/a | NCI | 0.67 | 0.49 |
| Phenylephrine | n/a | NCI | 0.63 | 0.91 |
| Theophylline | n/a | NCI | 0.37 | 0.53 |
| Morphine | n/a | NCI | 0.28 | 0.41 |
| Hydrocortisone | n/a | NCI | 0.23 | 0.34 |
| Atorvastatin | n/a | NCI | 0.15 | 0.21 |

down-regulates transforming growth factor (TGF)-α and EGFR levels in head and neck squamous cell carcinoma (46). Dimethyl sulfoxide (DMSO) has been associated with antiangiogenic effects (47), and theophylline enhances the sensitivity of lung cancer cells to cell death induction by other drugs (48, 49). Phenylephrine induces EGFR phosphorylation, which can be partially blocked by an EGFR TKI. In addition, statins like atorvastatin have been shown to enhance the tumor-inhibitory effects of various antitumor drugs, potentially reducing resistance in NSCLC patients (50, 51, 52, 53). The identification of these drug candidates holds promise for further investigations into EGFR–TKI resistance treatment strategies.

Despite the comprehensive approach of this study, it is not without limitations. The relatively small sample size due to challenges in obtaining in vivo experimental models for drug resistance is a constraint. In addition, individual variations among NSCLC patients, including socio-economic factors, disease severity, and duration, may influence result accuracy. Therefore, larger scale in vitro and in vivo experiments are warranted to validate the precise roles of hub genes in NSCLC and to guide future research efforts.

In our investigation, an analysis of gene expression was conducted between EGFR–TKI-sensitive and acquired drug-resistant samples, using data sourced from the GEO database. This rigorous analysis led to the identification of aberrant expression patterns within EGFR–TKI-resistant PDXs. Our study successfully pinpointed a total of 1,302 DEGs and highlighted 10 hub genes as pivotal players. The functional roles and pathways associated with these DEGs were substantiated through comprehensive Gene Ontology (GO) and KEGG enrichment analyses. Notably, our findings suggest that ITGAM may hold significant roles in the molecular pathogenesis of EGFR–TKI resistance. Moreover, the identified core genes and pathways exhibit promise as potential biomarkers, facilitating the detection and targeting of EGFR–TKI resistance in therapeutic interventions. In addition, the predicted drugs identified in our study offer the potential for use in combination with EGFR–TKIs, thus mitigating resistance in NSCLC patients and ultimately enhancing therapeutic efficacy. These discoveries contribute substantially to our comprehension of drug resistance mechanisms and the potential identification of targets to combat EGFR–TKI resistance. This, in turn, holds promise for the improvement of therapeutic outcomes in NSCLC patients. Nevertheless, it is imperative that further studies be conducted, encompassing a series of experimental investigations to validate our hypotheses and yield more precise correlation reports.

# Materials and Methods

### Microarray data

We initiated our study by querying the GEO database (http://www.ncbi.nlm.nih.gov/geo) using the search terms "NSCLC" and "EGFR-TKI resistant" (14). Specifically, we sought out PDX samples featuring EGFR mutations and exhibiting initial sensitivity to EGFR–TKI drugs, which were subsequently induced to develop acquired resistance within a PDX model. Our data analysis focused on two datasets, GSE64472 and GSE130160, for the evaluation of DEGs between EGFR–TKI-sensitive and resistant NSCLC samples. EGFR–TKI resistance was characterized by a threefold increase in tumor volume compared with pretreatment levels. The GSE64472 dataset included 3 EGFR–TKI-sensitive samples and 2 EGFR–TKI drug-resistant samples, whereas GSE130160 encompassed EGFR–TKI-sensitive samples and 1 EGFR–TKI-resistant sample. These datasets used the GPL6884 Illumina HumanWG-6 v3.0 expression beadchip and the GPL16791 Illumina HiSeq 2500 (*Homo sapiens*) platform.

### Screening of DEGs

The R package "limma" (http://www.bioconductor.org/) was used to normalize the data and execute differential expression analysis between EGFR–TKI-sensitive and acquired drug–resistant NSCLC tumor samples. DEGs were identified based on a false discovery rate–corrected $P$-value < 0.05 and |log$_2$-fold change (FC)| > 2. To ensure consistency, probe identification numbers were converted to gene symbols, with the maximum value selected as the gene expression value in cases where multiple probes corresponded to the same gene.

### Functional enrichment analysis of DEGs

To gain deeper insights into the functional roles and enriched pathways associated with the DEGs, we performed Gene Ontology (GO) analysis, encompassing biological processes, cellular components, and molecular functions, and KEGG pathway analysis. These analyses were conducted using the DAVID (https://david.ncifcrf.gov/) version 6.8 (12, 13, 20). Terms with $P$-values < 0.05 and representation by at least two enriched genes were considered statistically significant.

### PPI network construction and hub gene identification

The construction of a PPI network for the DEGs was carried out using the online STRING database (https://string-db.org). This step aimed to elucidate the key signaling pathways and cellular processes implicated in EGFR–TKI resistance in NSCLC (21). Subsequently, we visualized the network using Cytoscape Version 3.7.1. NetworkAnalyzer, a Cytoscape plug-in, facilitated the analysis of relationships among DEGs by computing network properties such as the clustering coefficient, node degree distribution, and shortest path length (23). The identification of candidate hub genes was performed with the cytoHubba plug-in, ranking genes based on their degree, closeness, and betweenness scores. The top 10 genes according to these scores were considered potential hub genes.

### Hub gene survival analysis

To assess the potential role of hub genes in drug resistance, we conducted DFS analysis for NSCLC patients using Kaplan–Meier curves from TCGA database (https://portal.gdc.cancer.gov/) (16). A log-rank test $P$-value < 0.05 was considered indicative of statistical significance. This analysis aimed to identify genes associated with the progression of drug resistance in NSCLC.

### Drug interaction prediction for EGFR–TKI resistance-specific key genes

To predict potential interactions between genes and drugs specific to EGFR–TKI resistance, we used the Drug Gene Interaction Database (DGIdb; www.dgidb.org). The EGFR–TKI resistance-specific key genes were input into DGIdb to identify targeted drugs with potential efficacy against EGFR–TKI resistance in NSCLC.

## Supplementary Information

## Author Contributions

L Zhu: conceptualization, resources, and writing—original draft.
S Gao: data curation, software, and formal analysis.
X Zhao: software, formal analysis, supervision, validation, visualization, and methodology.
Y Wang: conceptualization, resources, project administration, and writing—review and editing

## Conflict of Interest Statement

The authors declare that they have no conflict of interest.

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
