## [Reviewer comments · Life Science Alliance]

Life Science Alliance

Identification of biomarkers, pathways and therapeutic targets for EGFR-TKI resistance in NSCLC

leilei zhu, Shanshan Gao, Xianya Zhao, and Ying Wang

DOI: <https://doi.org/10.26508/lsa.202302110>

Corresponding author(s): Ying Wang, Anhui province Children Hospital

Review Timeline:

Submission Date:	2023-04-22
Editorial Decision:	2023-06-15
Revision Received:	2023-09-15
Editorial Decision:	2023-09-20
Revision Received:	2023-09-27
Accepted:	2023-09-28

Transaction Report:

June 15, 2023

Re: Life Science Alliance manuscript #LSA-2023-02110-T

Ying Wang
Anhui Children's Hospital

Dear Dr. Wang,

Thank you for submitting your manuscript entitled "Identification of biomarkers, pathways and potential therapeutic targets for EGFR-TKI resistance in non-small-cell lung cancer" to Life Science Alliance. The manuscript was assessed by an expert reviewer, whose comment is appended to this letter. We invite you to submit a revised manuscript addressing all the Reviewer comments.

Thank you for this interesting contribution to Life Science Alliance. We are looking forward to receiving your revised manuscript.

Sincerely,

B. MANUSCRIPT ORGANIZATION AND FORMATTING:

Reviewer #1 (Comments to the Authors (Required)):

The paper discussing EGFR-TKI resistance in lung cancer demonstrates the use of relevant models and sequencing data analysis to identify potential markers, investigate associated pathways, and explore sensitive targets. While the study provides valuable insights, it would benefit from more detailed bioinformatics analysis methods and additional experimental verification to strengthen its findings and provide robust evidence.

Major concerns:

1. Please clarify whether the data mentioned in the text refers to GSE64427 or GSE64472 to avoid confusion throughout the entire paper.
2. Figure 1c and d display data from two resistance groups and three sensitive groups from GSE64472, as well as two sensitive groups and one resistance group from GSE130160. However, the rationale behind this selection and the basis for these choices should be further explained. In fact, there are more groups in GSE64472 or GSE130160.
3. In order to identify genes most strongly associated with prognosis, the authors performed prognostic analysis after identifying hub genes related to TKI resistance. However, it is not clear why Disease-Free Survival (DFS) was used instead of Overall Survival (OS). Additionally, in Figure 3, although the P value of ITGAM is less than 0.05, the survival curves of the two groups with high or low expression of ITGAM are not completely separated. Can you explain it?
4. It is suggested to modify some graphics and statements to improve readability.
5. It is suggested to increase experimental verification of hub genes related to TKI resistance, such as drug sensitivity detection, etc.

Minor concerns:

Additional English language editing would be beneficial to this manuscript.

Dear Editors and Reviewer

Thank you for your letter and for the reviewer's comments concerning our manuscript entitled "Identification of biomarkers, pathways and potential therapeutic targets for EGFR-TKI resistance in non-small-cell lung cancer"(LSA-2023-02110-T). Those comments are all valuable and very helpful for revising and improving our paper, as well as the important guiding significance to our researches. We have studied comments carefully and have made correction which we hope meet with approval. Revised portion are marked in red in the paper. The main corrections in the paper and the responds to the reviewer's comments are as flowing:

Responds to the reviewer's comments

Reviewer #1 (Comments to the Authors (Required)):

The paper discussing EGFR-TKI resistance in lung cancer demonstrates the use of relevant models and sequencing data analysis to identify potential markers, investigate associated pathways, and explore sensitive targets. While the study provides valuable insights, it would benefit from more detailed bioinformatics analysis methods and additional experimental verification to strengthen its findings and provide robust evidence.

Major concerns:

1. Please clarify whether the data mentioned in the text refers to GSE64427 or GSE64472 to avoid confusion throughout the entire paper.
2. Figure 1c and d display data from two resistance groups and three sensitive groups from GSE64472, as well as two sensitive groups and one resistance group from GSE130160. However, the rationale behind this selection and the basis for these choices should be further explained. In fact, there are more groups in GSE64472 or GSE130160.
3. In order to identify genes most strongly associated with prognosis, the authors performed prognostic analysis after identifying hub genes related to TKI resistance. However, it is not clear why Disease-Free Survival (DFS) was used instead of Overall Survival (OS). Additionally, in Figure 3, although the P value of ITGAM is less than 0.05, the survival curves of the two groups with high or low expression of ITGAM are not completely separated. Can you explain it?
4. It is suggested to modify some graphics and statements to improve readability.
5. It is suggested to increase experimental verification of hub genes related to TKI resistance, such as drug sensitivity detection, etc.

Reviewers' comments:

Reviewer #1 (Comments to the Authors (Required)):

The paper discussing EGFR-TKI resistance in lung cancer demonstrates the use of relevant models and sequencing data analysis to identify potential markers, investigate associated pathways, and explore sensitive targets. While the study provides valuable insights, it would benefit from more detailed bioinformatics analysis methods and additional experimental verification to strengthen its findings and provide robust evidence.

Author response: Thank you. We appreciate your comments and suggestions.

Major concerns:

1. Please clarify whether the data mentioned in the text refers to GSE64427 or GSE64472 to avoid confusion throughout the entire paper.

Author response: Thank you. We apologize for causing confusion. We have revised the confusion “64427” has been replaced by “64472” in the revised manuscript.

2. Figure 1c and d display data from two resistance groups and three sensitive groups from GSE64472, as well as two sensitive groups and one resistance group from GSE130160. However, the rationale behind this selection and the basis for these choices should be further explained. In fact, there are more groups in GSE64472 or GSE130160.

Author response: Thank you. We appreciate this comment. We appreciate your interest in the sources of sample, and thank you for your kind suggestion. We apologize for causing confusion. In the studies, our description of human data selection not sufficiently clear. Our study investigates the mechanism of drug resistance in the application of epidermal growth factor receptor tyrosine kinase inhibitors (EGFR-TKIs) in NSCLC patients. Histological analyses revealed that PDXs showed a histology similar to that of patients' surgically resected tumors (SRTs), In order to make the results more reliable, we collected samples of Patient derived xenograft (PDX) models as the research subjects . In the GSE130160 study, they successfully established ten PDXs, including three adenocarcinoma (AD), six squamous cell carcinoma (SQ) and one large cell carcinoma (LA), from 30 patients with non-small cell lung cancer (NSCLC) (18 AD, 10 SQ, and 2 LA), mainly in SHO mice (Crj:SHO-PrkdcscidHrhr). Two out of three PDXs with AD histology had EGFR mutations (L858R or exon19 deletion) and were sensitive to EGFR tyrosine kinase inhibitors (EGFR-TKIs), such as gefitinib and osimertinib. Fortunately, in one of the two PDXs with an EGFR mutation, osimertinib resistance was induced that was associated with epithelial-to-mesenchymal transition. Therefore, we included these two sensitive samples with the EGFR mutation, and one drug-resistant sample for subsequent research. In the GSE64472 study, they successfully established three PDXs of human NSCLC, Overall design Murine models of human NSCLC were generated and targeted inhibition studies were performed using AZD2171 (cediranib) and ZD6474 (vandetanib). Cediranib is a TKI through the VEGFR mechanism, while vandetanib is a TKI that acts on both EGFR and VEGFR pathways. We selected three vandetanib sensitive samples and two drug-resistant samples from the GSE64472 study for our subsequent research. Thank you very much for your understanding and support.

1. In order to identify genes most strongly associated with prognosis, the authors performed prognostic analysis after identifying hub genes related to TKI resistance. However, it is not clear why Disease-Free Survival (DFS) was used instead of Overall Survival (OS)

Author response: Thank you for your insightful suggestion. We apologize for causing confusion. At present, tyrosine kinase inhibitor (TKI) treatment is the first-line therapy

for some tumors. Acquired resistance represents a bottleneck to molecularly targeted therapies such as epidermal growth factor receptor (EGFR) TKI treatment in lung cancer. A deeper understanding of resistance mechanisms can provide insights into this phenomenon and help to develop additional therapeutic strategies to overcome or delay resistance. The manifestation of drug resistance in tumors is tumor growth and metastasis. The clinical manifestation is the progression and deterioration of the disease. Disease-free survival (DFS), refers to the time from randomization to disease recurrence or death due to disease progression. For the above considerations, we chose DFS instead of OS as the prognostic observation indicator. Thank you very much for your understanding and support.

Additionally, in Figure 3, although the P value of ITGAM is less than 0.05, the survival curves of the two groups with high or low expression of ITGAM are not completely separated. Can you explain it?

Author response: Thank you for your insightful suggestion. In this study, We used the Kaplan–Meier Plotter database to explore how these hub genes were related to NSCLC patient DFS, with data sourced from GEO, TGA, and TCGA. Due to the fact that the cases come from different studies, the follow-up time of the cases (after the cut-off point) varies. For this reason, the survival curves of the two groups with high or low expression of ITGAM intersect at the end. Thank you very much for your understanding and support.

4. It is suggested to modify some graphics and statements to improve readability.

Author response: Thank you very much for this valuable suggestion. The sentences in the main text have been modified to improve readability. Thank you very much for your understanding and support.

5. It is suggested to increase experimental verification of hub genes related to TKI resistance, such as drug sensitivity detection, etc.

Author response: Thank you for this valuable suggestion. Because of the time constraints, we have not conducted further experimental verification, and hope that you can understand our position and support our current inability to proceed. We appreciate this important point from the reviewer and have provided additional discussion relating to this point (see Discussion, Page 9, Line 257-261). Thank you very much for your understanding and support.

Minor concerns:

Additional English language editing would be beneficial to this manuscript.

Author response: Thank you for this valuable suggestion. This manuscript has been language edited by Elsevier Language Editing Service. Thank you very much for your understanding and support.

September 20, 2023

RE: Life Science Alliance Manuscript #LSA-2023-02110-TR

Ms. Ying Wang
Anhui province Children Hospital
39 Wangjiang East Road, Baohe District, Hefei City, Anhui Province
Hefei 230001
China

Dear Dr. Wang,

Thank you for submitting your revised manuscript entitled "Identification of biomarkers, pathways and therapeutic targets for EGFR-TKI resistance in NSCLC". We would be happy to publish your paper in Life Science Alliance pending final revisions necessary to meet our formatting guidelines.

- please upload your figures as single files
- please remove your figures from the main manuscript file
- please add an Abstract and a Summary Blurb/Alternate Abstract to our system
- please note that the abstract should be a single paragraph not exceeding 175 words
- please add a Category for your manuscript in our system
- please add the Twitter handle of your host institute/organization as well as your own or/and one of the authors in our system
- please note that the titles in the system and on the manuscript file must match
- please place your figure legends after the reference section
- figure 4 is wrongly labeled as Figure 3; please correct
- please incorporate any points from the Conclusion section into the Discussion; we only allow a Discussion section

A. FINAL FILES:

B. MANUSCRIPT ORGANIZATION AND FORMATTING:

Sincerely,

Reviewer #1 (Comments to the Authors (Required)):

I have no additional inquiries regarding the revised manuscript. The authors have satisfactorily addressed all of my previous questions.

September 28, 2023

RE: Life Science Alliance Manuscript #LSA-2023-02110-TRR

Ms. Ying Wang
Anhui province Children Hospital
39 Wangjiang East Road, Baohe District, Hefei City, Anhui Province
Hefei 230001
China

Dear Dr. Wang,

Thank you for submitting your Resource entitled "Identification of biomarkers, pathways and therapeutic targets for EGFR-TKI resistance in NSCLC". It is a pleasure to let you know that your manuscript is now accepted for publication in Life Science Alliance. Congratulations on this interesting work.

DISTRIBUTION OF MATERIALS:

Again, congratulations on a very nice paper. I hope you found the review process to be constructive and are pleased with how the manuscript was handled editorially. We look forward to future exciting submissions from your lab.

Sincerely,
